# Crotalphine Modulates Microglia M1/M2 Phenotypes and Induces Spinal Analgesia Mediated by Opioid-Cannabinoid Systems

**DOI:** 10.3390/ijms231911571

**Published:** 2022-09-30

**Authors:** Flavia S. R. Lopes, Aline C. Giardini, Morena B. Sant’Anna, Louise F. Kimura, Michelle C. Bufalo, Hugo Vigerelli, Vanessa O. Zambelli, Gisele Picolo

**Affiliations:** 1Laboratory of Pain and Signaling, Butantan Institute, São Paulo 05503-900, Brazil; 2Center of Excellence in New Target Discovery (CENTD), Instituto Butantan, São Paulo 05503-900, Brazil

**Keywords:** chronic pain, glial cells, neuroimmune modulation, M2 polarization, endogenous opioid release, neuropathic pain

## Abstract

Pain is a worldwide public health problem and its treatment is still a challenge since clinically available drugs do not completely reverse chronic painful states or induce undesirable effects. Crotalphine is a 14 amino acids synthetic peptide that induces a potent and long-lasting analgesic effect on acute and chronic pain models, peripherally mediated by the endogenous release of dynorphin A and the desensitization of the transient receptor potential ankyrin 1 (TRPA1) receptor. However, the effects of crotalphine on the central nervous system (CNS) and the signaling pathway have not been investigated. Thus, the central effect of crotalphine was evaluated on the partial sciatic nerve ligation (PSNL)-induced chronic neuropathic pain model. Crotalphine (100 µg/kg, p.o.)-induced analgesia on the 14th day after surgery lasting up to 24 h after administration. This effect was prevented by intrathecal administration of CB1 (AM251) or CB2 (AM630) cannabinoid receptor antagonists. Besides that, crotalphine-induced analgesia was reversed by CTOP, nor-BNI, and naltrindole, antagonists of *mu*, *kappa*, and *delta*-opioid receptors, respectively, and also by the specific antibodies for β-endorphin, dynorphin-A, and met-enkephalin. Likewise, the analgesic effect of crotalphine was blocked by the intrathecal administration of minocycline, an inhibitor of microglial activation and proliferation. Additionally, crotalphine decreased the PSNL-induced IL-6 release in the spinal cord. Importantly, in vitro, crotalphine inhibited LPS-induced CD86 expression and upregulated CD206 expression in BV-2 cells, demonstrating a polarization of microglial cells towards the M2 phenotype. These results demonstrated that crotalphine, besides activating opioid and cannabinoid analgesic systems, impairs central neuroinflammation, confirming the neuromodulatory mechanism involved in the crotalphine analgesic effect.

## 1. Introduction

Studies have estimated that 20% of the adult worldwide population suffers from different types of pain [1]. Despite the advances in the research in this field, the treatment for chronic pain still presents great challenges, since even with the administration of analgesic drugs like opioids, alone or in combination with other classes of drugs such as antidepressants and antiepileptics, many patients report an incomplete reversal of the painful state and/or undesirable adverse effects [2]. One of the reasons related to the complexity in the treatment of pain concerns its origin, which can result from an inflammatory process and/or nerve damage. Both processes modify the harmful stimulus transmission and the functionality of the sensory system (nociception) as well as the neuroimmune interface in both peripheral (PNS) and central (CNS) nervous systems [3]. This altered condition at both PNS and CNS not only changes the neuronal network but also involves interactions between neurons and other cells, including endothelial cells, macrophages, T cells, and glial cells, mediated by pro-inflammatory cytokines and chemokines release [4,5]. 

Chronic pain development involves important modifications in the spinal cord including glial cell activation [6,7]. Specifically, microglial cells play a significant role in central sensitization and neuropathic pain development by changing the synaptic function and by releasing a variety of pro-inflammatory cytokines with different degrees of proliferation or hypertrophy in these cells [8,9]. However, an emerging role of microglia in the resolution of pain has also been stated, in which anti-inflammatory cytokines and specialized pro-resolving mediators are released by these cells, including resolvins and protectins, which may act as potent neuromodulators [10]. In addition, activated microglia is also responsible for the production and release of analgesic compounds, such as endorphins and endocannabinoids, both in vivo and in vitro, thus regulating pain transmission [11,12]. Both opioid and cannabinoid compounds are known to be effective analgesics by reducing the inflammation and excitability of neuronal cells leading to the decreased transmission of nerve impulses alongside the inhibition of neurotransmitter release [13,14,15]. These systems are widely distributed in the PNS as well as in the CNS and can be endogenously or exogenously activated [16]. In addition, they can interact with each other directly via heterodimerization of receptors, or indirectly through the release of endogenous ligands that, in turn, can activate their receptors [17]. In addition, the activation of the cannabinoid system for pain control has allowed, when compared to chronic opioid therapy, the use of lower doses with fewer side effects. Thus, the cross-talk between these systems could be a feasible alternative to be explored for chronic pain control [18]. 

Crotalphine is a synthetic peptide produced according to the sequence of the natural analgesic compound firstly identified in *Crotalus durissus terrificus* snake venom. When orally administered, crotalphine induces potent and long-lasting antinociceptive effects in both acute and chronic pain models, including chronic constriction injury of the sciatic nerve and bone cancer-induced pain [19,20,21]. It was demonstrated that, peripherally, its analgesic effect occurs by the release of endogenous opioids [19,20], particularly dynorphin A, this release being dependent on the previous activation of cannabinoid type 2 (CB2) receptors [22]. It was also demonstrated that crotalphine induces the desensitization of TRPA1 (transient receptor potential ankyrin 1) receptors [23], members of the TRP superfamily. Importantly, crotalphine does not induce side effects commonly observed for opioids or other analgesic drugs, such as tolerance (even after chronic administration—every 3 days for 75 days), or alteration in general and spontaneous motor activity [20]. Crotalphine does not interfere with the viability of DRG cells [24] or of human sensory-like neurons SH-SY5Y (data not published) and does not interfere with the body weight of rodents [25]. However, no study has yet investigated crotalphine effects on the CNS. 

Thus, the spinal mechanisms involved in crotalphine analgesic effects on a chronic neuropathic pain model are currently being investigated. It was observed that, besides activating opioid and cannabinoid systems, crotalphine modulates central neuroinflammation induced by neuropathy, in a pathway that involves glial cells as key players, confirming the neuroimmune role of this peptide in pain control at the spinal cord level. 

## 2. Results

### 2.1. Crotalphine Induces Dose and Time-Dependent Analgesia in the PSNL Model of Neuropathic Pain

In order to evaluate the effect of crotalphine in the acute or chronic phases of hypernociception induced by PSNL, it was first characterized as the course of pain induced by the model. The mechanical hypernociception was determined using the *von* Frey filaments test, according to item 4.5. Mice under PSNL showed a marked decrease in the nociceptive threshold that was already observed 1 day after surgery lasting for up to 21 days (Figure 1A). Sham animals presented a slight decrease in their pain threshold that was restored to basal level on the 3rd day after surgery (Figure 1A). Therefore, the analgesic effect of crotalphine in the acute inflammatory phase of the model was determined on the 3rd day after surgery whereas the effect of the treatment in the chronic neuropathic phase was evaluated on the 14th day after surgery. PSNL group was orally treated (p.o) with 5, 50, or 100 µg/kg of crotalphine, based on previous results [22], or with vehicle (saline). Crotalphine administered on the 3rd day after surgery reduced the PSNL-induced hypernociception for up to 24 h at 5 and 50 µg/kg and up to 48 h at the dose of 100 µg/kg (Figure 1B). When crotalphine was administered to PSNL mice on the 14th day after surgery, the period in which neuropathy is already installed [26], the doses of 5 and 50 µg/kg induced analgesia 1 h after administration, but was no longer observed 24 h after treatment. In contrast, the PSNL group treated with the 100 µg/kg of crotalphine exhibited an analgesic effect observed 1 h after treatment, lasting partially up to 24 h after administration (Figure 1C). Therefore, the dose of 100 µg/kg of crotalphine was chosen for the next experiments. The mechanisms involved in crotalphine-induced analgesia were evaluated in the chronic phase of pain. Importantly, the control of the acute phase of the model did not affect the establishment of the chronic phase of the neuropathy (Appendix A, Analgesia induced by crotalphine in the acute phase does not prevent the development of the chronic pain).

### 2.2. Spinal Cannabinoid Receptors Mediate the Analgesic Effect Induced by Crotalphine

Since it was previously observed that the peripheral analgesic effect of crotalphine involves the activation of the cannabinoid pathway through the activation of peripheral CB2 receptors [22], the contribution of central receptors to the crotalphine analgesic effect was investigated. The results demonstrated that the intrathecal administration of AM251, a CB1 receptor antagonist (Figure 2A), or AM630, a CB2 receptor antagonist (Figure 2B), blocked the analgesic effect induced by crotalphine (p.o.), confirming the involvement of both subtypes of cannabinoid receptors, located at the spinal cord, in the crotalphine-induced analgesia. 

### 2.3. Spinal Opioid Pathway Is Involved in the Analgesic Effect Induced by Crotalphine

Considering that the peripheral effect of crotalphine on the acute pain model involves the endogenous release of dynorphin A and that this release depends on the previous activation of CB2 receptors [22], it was next investigated whether the central spinal opioid system would also be involved in crotalphine effect. On the 14th day after PSNL, mice were intrathecally treated with *mu*, *kappa*, or *delta-opioid* receptor antagonists, and after 10 min animals were orally treated with crotalphine. The nociceptive threshold was assessed 1 h after the administration of crotalphine. Treatment with CTOP (Figure 3A), nor-BNI (Figure 3B), or naltrindole (Figure 3C) abolished the analgesic effect of crotalphine. 

Moreover, specific antibodies delivered intrathecally for β-endorphin (Figure 3D), dynorphin-A (Figure 3E), and met-enkephalin (Figure 3F) also blocked the analgesic effect of crotalphine, suggesting that, differently from observed at the periphery, this effect involves the central release of endogenous opioids which activates *mu, kappa,* and *delta* receptors, respectively.

### 2.4. Participation of Microglia in the Analgesic Effect of Crotalphine

Considering that (1) crotalphine, when evaluated on the macrophages functions, induces a dual effect over these cells, inhibiting some functions such as phagocytosis, H_2_O_2_ release, nitric oxide production, and TNF-α secretion and, on the other hand, stimulating the secretion of IL-1β and modulating the secretion of IL-6 (stimulating or inhibiting dependent on the cell state—resident or activated) (unpublished data); (2) microglia are macrophage-like cells in the CNS [27]; (3) microglial cells are key players in central sensitization modulation, releasing both neuroinflammatory mediators, contributing to its development, as wells as anti-inflammatory and analgesic compounds, being able to downregulate central activity, and (4) microglia are pivotal cells for endogenous opioids peptides and endocannabinoid release [10], our next step was to verify the participation of these cells in the analgesia induced by crotalphine. On the 14th day after PSNL, mice received intrathecally minocycline, a semisynthetic second-generation tetracycline, which acts as a potent inhibitor of microglial activation and proliferation [28], 30 min before crotalphine. The results showed that minocycline reversed crotalphine-induced analgesia, confirming the involvement of microglia in this analgesic effect (Figure 4).

### 2.5. Quantification of Interleukin 6 Levels in the Spinal Cord of Mice with Neuropathy after Treatment with Crotalphine

It is widely described that, as peripheral macrophages, microglia may assume neural-specific phenotypes according to different stimuli, polarizing to an M1 phenotype, in which they express and release pro-inflammatory cytokines and contribute to neuropathic pain or an M2 phenotype for the resolution of inflammation and tissue repair [29]. Because the analgesic effect of crotalphine involves the release of endogenous analgesic peptides, and considering that microglia is an important central source of these peptides [30], and that, once microglia is inhibited, the analgesic effect of crotalphine is lost, we sought to assess whether crotalphine could interfere with the levels of important cytokines released by activated microglia such as IL-6. Mice with neuropathy (14th day after PSNL) were treated with crotalphine and one hour later the ipsilateral spinal cord was collected for IL-6 quantification. Our data identified that mice treated with crotalphine, where analgesia was observed (Figure 5A) showed lower levels of IL-6 when compared with PSNL mice (Figure 5B), reinforcing the effect of the peptide on immune cells.

### 2.6. Effect of Crotalphine on LPS-Induced BV2 Cells Polarization to M1 or M2 Phenotypes

Considering the results above, which suggest that microglia is a key player in the antinociceptive effect of crotalphine, the effect of this peptide on M1 and M2 phenotype polarization using the BV-2 mouse microglial cell line was checked next. The levels of CD86 and CD206, which are standard M1 and M2 markers, respectively, [30,31] were subsequently evaluated by flow cytometry assay. Lipopolysaccharide (LPS) was used to promote M1 phenotype polarization since it induces a neuroinflammatory state of the cell via interaction with microglial membrane receptors [32,33].

Cells were assigned into the following groups: control (medium), LPS (100 ng/mg), LPS + 0.1 µM crotalphine, LPS + 1 µM crotalphine, and LPS + 10 µM crotalphine. For the LPS + crotalphine group, the cells were pretreated with LPS (100 ng/mg) for 24 h, the medium was removed, and then the cells were treated with different concentrations of crotalphine (0.1 µM, 1 µM, and 10 µM) for 24 h.

As shown in Figure 6A, the expression of CD86 on BV-2 cells was increased by LPS, confirming the polarization of microglial cells to the M1 state. The LPS-induced CD86 expression was markedly downregulated by crotalphine. In addition, crotalphine increased the expression of CD206 in BV2 cells compared with both LPS and control groups. The flow cytometry histograms representing the surface expression of the CD86 and CD206 molecules in BV-2 cells are represented in the Appendix A (Appendix A, Representative histograms of surface expression of CD86 and CD206 molecules on BV-2 cells). These results confirm the potential of crotalphine to enhance the transition from M1 to M2 phenotype polarization.

## 3. Discussion

Pain represents an important clinical, social, and economic problem affecting patients of all ages, with prevalence estimates ranging from 1.0% to over 60.0% monthly. In addition, pain is one of the main causes of disability worldwide presenting a negative impact on the quality of life and affecting more Americans than diabetes, heart disease, and cancer combined [34]. Given this information, a more precise understanding of this complex phenomenon is needed to develop more effective management strategies.

Crotalphine is a 14 amino acid peptide that induces long-lasting analgesic effects in acute and chronic pain models in low doses and by different routes of administration [19,20,21]. Peripherally, this effect is mediated by the activation of peripheral *kappa*- and/or *delta*-opioid receptors and involves the activation of cannabinoid CB2 receptors and the subsequent release of dynorphin A, the endogenous agonist of *kappa* opioid receptors, suggesting an interaction between the opioid and cannabinoid systems [22]. However, although much is known about the mechanisms involved in its analgesic effect [20,23,35], the mechanism of crotalphine action on the central nervous system remains unknown. The results currently obtained highlighted the modulatory effect of crotalphine on glial cells, shifting microglia from M1 to M2 state, and the phenotype of microglia associated with the control of neuroinflammation. We also demonstrated that the effect of crotalphine involves the opioid-cannabinoid systems with endogenous opioid release. Since glial cells are important sources of endogenous opioid peptides and also a target for opioid and cannabinoid action, we point out the action of crotalphine over glial cells as an important step for chronic pain control.

The partial sciatic nerve injury (PSNL) model of neuropathy currently used induces numerous anatomical, neurochemical, and gene and protein expression changes, at the injured nerve, in the dorsal root ganglion (DRG), and in the spinal cord already described elsewhere [5,26,36]. These changes result in the development of neuropathic pain symptoms, including hypersensitivity to both mechanical and thermal stimuli [5,26,36], associated with an increase in the inflammatory response, immune cell reactions, and the activation of glial cells [37]. In general, excessive and prolonged activation of these cells can result in pathological inflammation that contributes to the progression of chronic diseases [38].

In the model currently used, crotalphine orally administered in the acute phase (3rd day after surgery) induced analgesia observed for up to 2 days, but this treatment did not interfere with the onset of neuropathy. In the chronic phase, partial reversion of PSNL-induced pain behavior was observed for up to 24 h after administration. Even though this effect differs from those previously observed in the CCI model of neuropathic pain, where crotalphine analgesia was observed for longer periods [20], the peptide-induced analgesic effect observed herein still lasts longer than the most clinically available analgesic drugs. A possible explanation for these dissimilarities may be a different neuroimmune interface between the PSNL and CCI models, such as the profile and the time of cell activation, the release of cytokines, and chemokines in different tissues, and kinase signaling pathway activation [39].

As already mentioned, peripherally, crotalphine-induced analgesia is mediated by peripheral *kappa* and cannabinoid CB2 receptors. The activation of these receptors induced by crotalphine, at the periphery, was demonstrated using conformation state sensitive antibodies against activated *kappa* opioid receptors and activated CB2 cannabinoid receptors, in the skin tissue of the paw, in the presence or not of a specific CB2 antagonist [22], confirming the ability of crotalphine to stimulate both systems. Currently, our data revealed that the central effect of crotalphine involves the participation of *mu*, *kappa*, and *delta*-opioid receptors, and their endogenous ligands β-endorphin, met-enkephalin, and dynorphin A release, respectively, in addition to CB1 and CB2 cannabinoid receptors. Preliminary studies indicate that crotalphine does not directly activate opioid or cannabinoid receptors since the peptide is not able to displace the labeled naloxone ([3H] naloxone) or even the non-selective labeled agonist of CB1 and CB2 receptors ([3H] CP55,490 in binding studies (unpublished data), suggesting that this effect may be due to an indirect effect or it occurs via its sub-products or metabolites. Then, although our data confirm the involvement of both opioid and cannabinoid systems in the analgesia induced by crotalphine, we cannot confirm if the activation of these receptors by crotalphine occurs directly or indirectly. This evaluation is necessary and will be further investigated. 

It is already described that both opioid and cannabinoid systems share many characteristics when activated, triggering similar responses including antinociception, suggesting common pathways of action. In addition, these two types of receptors are found in regions of the brain known to modulate the analgesic process by acting synergistically within the same cell or neuronal circuit to induce analgesia. Likewise, the modulation of one receptor may cause changes in the activity of the other and vice-versa, indicating a robust cross-talk between these two systems [40]. At the CNS, both endogenous opioids and endocannabinoids may be released by microglia [10,12].

Microglia acts as principal innate immune cells in the central nervous system (CNS) [41]. Spinal microglia plays a crucial role in the initiation and development of chronic pain after peripheral inflammation or injury, mainly due to the release of several mediators such as neurotrophins and pro-inflammatory cytokines, which leads to central sensitization [12]. In contrast, active microglia can also be in a “protective state” induced by anti-inflammatory cytokines, which activate a cascade of tissue repair mechanisms, leading to the release of several microglial analgesics and anti-inflammatory molecules such as endogenous opioids, anti-inflammatory cytokines, pro-resolution mediators including resolvins (resolution-phase interaction products), protectins/neuroprotectins, and maresins (macrophage mediators in resolving inflammation) [10,12,42]. Importantly, an in vitro study showed that the activation of microglial cannabinoid receptor type 2 (CB2) acts as a trigger for the anti-inflammatory phenotype of microglia by inducing MAP kinase phosphatase-3 expression which selectively inhibits p-ERK, resulting in a reduction in TNFα expression and microglial motility. Authors suggested that the reduction in pro-inflammatory microglia migration would reduce the source of pro-analgesic mediators, such as TNFα, preventing neuronal sensitization and hence alleviating pain behavior [43]. Based on that, we hypothesized that the spinal glia is a fundamental target for the crotalphine effect. In fact, our data demonstrated that the treatment with minocycline, a non-specific microglial inhibitor, blocked the analgesic effect of crotalphine. Data from the literature have shown that the production and release of β-endorphin in the spinal cord of rats and cultured microglia stimulated by LPS were completely blocked by minocycline [12,44]. Minocycline also suppressed β-endorphin release stimulated by a glucagon-like peptide-1 (GLP-1) receptor agonist [12] and by thalidomide, an immunomodulatory agent which induces analgesia through the upregulation of spinal microglia and β-endorphin expression [8]. Moreover, it is suggested that the microglial p38 mitogen-activated protein kinase (p38-MAPK) pathway is involved in the β-endorphin release and that minocycline prevented this release attenuating p38 activation [45,46]. Interestingly, our group has previously demonstrated that crotalphine increases the activation of spinal MAPK and that its analgesic effect is dependent on the activation of ERK1/2 and JNK MAPK via the PKCζ signaling pathway [24]. Therefore, we can suggest that the effect of crotalphine on spinal microglial cells is a key step for antinociceptive effect, through the release of endogenous opioid peptides. 

To reinforce our hypothesis that microglia cells are involved in the analgesic effect of crotalphine acting towards a “protective state”, we evaluated (1) the release of interleukin-6 (IL-6), an important cytokine released by glia cells under inflammatory conditions, widely increased in the PSNL model, which modulates the nociceptive function [6,7]; and (2) the ability of crotalphine to polarize the transition from M1 to M2 phenotypes in a BV-2 microglial cells line. In vitro studies have shown that the agonism of CB2 by AM1241 reduces IL-6 levels by murine N9 microglial cells and also shifts the M1 microglial phenotype to M2 when exposed to inflammatory mediators [47]. Moreover, the CB2 gene deletion from murine primary microglial cells also reduced the IL-6 release induced by inflammatory agents such as LPS and IFNγ [48]. Likewise, in an animal model of traumatic brain injury, the agonism of CB2R (GP1a) attenuated pro-inflammatory M1 macrophage polarization, increased anti-inflammatory M2 polarization, and reduced IL-6 expression [49]. Our results demonstrated, in vivo, that the PSNL increased the spinal IL-6 content which was prevented by crotalphine treatment. Moreover, in vitro, crotalphine downregulated LPS-induced CD86 expression, a marker of the M1 state, and increased the expression of CD206, a marker of the M2 state, in BV2 cells, reinforcing the potential of crotalphine in modulating microglial states and hence confirming that crotalphine induces a microglial-mediated protective neuroimmune response. 

Importantly, although the results highlighted that glial cell modulation is a key step for the central antinociceptive effect of crotalphine, the concentration of the peptide in the spinal cord of PSNL mice after oral administration was not investigated. The blood–brain barrier (BBB) is a selective and permeable physical barrier with the role of protecting the central nervous system (CNS) environment. BBB restricts the access of a wide range of compounds into the CNS, including toxins, pathogens, antibodies, and immune cells, and simultaneously prevents the entrance of many therapeutic drugs [50,51]. 

In general, hydrophobic molecules can cross the BBB more easily than hydrophilic drugs. As a first step to overcome this challenge, to predict the crotalphine physiochemical properties and its blood–brain barrier-penetrating potential, we used analyzing tools to determine the aliphatic index and grand average of hydropathicity (GRAVY). The blood-brain barrier-penetrating potential, analyzed by computational models, indicates that crotalphine is not predicted to easily penetrate the blood–brain barrier, with a probability of only 0.36 and 0.12 (Appendix A, Analysing tool for synthetic crotalphine physiochemical properties prediction and blood-brain barrier penetrating potential). However, it is known that several disorders and pathological conditions may dysregulate the BBB integrity and increase the permeability, such as stroke, brain tumors, meningoencephalitis, multiple sclerosis, and neuroinflammation [50,52,53], which allows the transmigration of activated cells and the entrance of several compounds into the CNS [54]. The manipulation of molecular transport through BBB for the flow of solutes into the brain has been widely described [53,55,56]. Then, we cannot confirm whether the modulation of glial cell phenotypes occurs through a direct action of crotalphine in the CNS or indirectly, via its sub-products, its signaling pathway, and/or its metabolites. Our current results demonstrated, in BV-2 cells, the ability of crotalphine to directly modulate glial cell polarization. Nevertheless, we also demonstrated that crotalphine induces the peripheral release of dynorphin A [22] which would also contribute to glial cell polarization since it was demonstrated that *kappa* opioid receptors activation by dynorphin promotes microglial polarization toward the M2 phenotype via TLR4/NF-κB pathway [57]. So, it is a limitation of this study that will be further investigated.

## 4. Materials and Methods

### 4.1. Animals

Male C57BL/6 mice (19–22 g, aged 8–12 weeks) from the animal facility of Butantan Institute (Sao Paulo, Brazil) were used throughout this study. Animals were housed four to five per cage in a temperature-controlled (21 ± 2 °C) and light-controlled (12/12 h light/dark cycle) room with food and water ad libitum. Mice were adapted to these conditions at least 3 days before the beginning of the experiments. All behavioral tests were performed between 09:00 and 16:00 h.

### 4.2. Neuropathic Pain Induction—Partial Sciatic Nerve Ligation (PSNL) Model

For chronic neuropathic pain induction, the partial sciatic nerve ligation model was used, since it mimics important symptoms observed in patients suffering from neuropathy of inflammatory origin [58]. The partial sciatic nerve ligation (PSNL) surgery was performed as previously described by Malmberg and Basbaum [26] with slight modifications. Briefly, mice were anesthetized (1.5% isoflurane in oxygen) and the left sciatic nerve was exposed at the medial part of the thigh and approximately 1/3 to 1/2 of the diameter was tightly tied with 8.0 silk suture thread (8.0, Bioline, BR). Next, the skin was sutured with 4.0 silk suture thread (4-0, Bioline, BR). Control animals (sham group) also underwent surgery, but the sciatic nerve was only exposed, with no manipulation, followed by a skin suture. Naïve mice were also used as a control of surgery. The nociceptive threshold was assessed before and after surgery at specific time points. 

### 4.3. Crotalphine Treatment

Crotalphine (Proteimax, São Paulo, Brazil—Lot: P170717-HS595874) was diluted in distilled water and administered at 5, 50, or 100 μg/kg doses (in sterile saline, 200 µL p.o.) by oral route (p.o., using a gavage needle of 1.0 × 25.8 mm, #CA2800, Ciencor, São Paulo, Brazil). 

### 4.4. Pharmacological Treatments 

For in vivo signaling pathways determination, the following compounds were used: 

*Opioid signaling pathway*: CTOP (#P5296, Sigma Aldrich, St. Louis, MO, USA), a *mu* opioid receptor antagonist, was administered at 150 ng/10 µL (in sterile saline, i.t.) [59]; nor-BNI (#N1771, Sigma Aldrich, St. Louis, MO, USA), a *kappa* opioid receptor antagonist, was administered at 60 μg/10 µL (in sterile saline, i.t.) [60]; naltrindole (#N115, Sigma Aldrich, St. Louis, MO, USA), a *delta* opioid receptor antagonist, was administered at 10 μg/10 µL (in sterile saline, i.t.) [61]; anti-met-enkephalin Antibody (#T-4293, Peninsula Laboratories International, San Carlos, CA, USA) was administered at 20 μg/10 µL (in distilled water, i.t.); Anti-β-endorphin antibody (#T-4044, Peninsula Laboratories International, San Carlos, CA, USA) was administered at 10 μg/10 µL (in sterile saline, i.t.); anti-dinorphin A antibody (#T-4267, Peninsula Laboratories International, San Carlos, CA, USA) was administered at 10 μg/10 µL (in sterile saline i.t.).

*Cannabinoid receptors:* AM251 (#A6226, Sigma Aldrich, St. Louis, MO, USA), a CB1 receptor antagonist, was administered at 10 μg/10 µL (in sterile saline, i.t.) [62]; AM630 (#SML0327, Sigma Aldrich, St. Louis, MO, USA), a CB2 receptor antagonist, was administered at 2 μg/10 µL (in sterile saline, i.t.) [63].

*Glial cell:* Minocycline, a semisynthetic second-generation tetracycline, which acts as a potent inhibitor of microglial activation and proliferation [28], was administered at 10 nM/10 µL (in sterile saline, i.t.). 

These compounds were administered by the intrathecal route (i.t.). Mice were kept under anesthesia (1.5% Isoflurane in oxygen) and trichotomy was performed in the dorsal region. An insulin syringe (8 × 0.3 mm—30 G, BD Ultra-Fine II, Becton Dickinson and Company, East Rutherford, NJ, USA) was inserted between the L5-L6 vertebrae, and the solution was administered (10 µL) only when animals showed an involuntary tail flick, which indicated the correct insertion [64].

### 4.5. Von Frey Filaments Test for Hypernociception Determination

Mice were placed on a metal grid (0.8 × 0.8 cm), elevated 30 cm from the surface of the bench, on which an acrylic box (8 × 8 × 18 cm) divided into ten equal compartments was placed. Before testing, animals remained in the experimental environment (room and apparatus) for approximately 20 min until exploratory activity decreased. The mechanical nociceptive threshold was assessed using the *von* Frey filaments by applying a mild and constant stimulus to the central region of the ipsilateral paw, until a withdrawal response or flinch of the hind paw was observed. The von Frey filaments test consists of a logarithmic series of 8 nylon filaments (Anesthesiometer Semmer-Weinster, Stoelting Co., Wood Dale, IL, USA) with progressive force variation. The value of each filament is given in log10 (milligrams), ranging from 1.65 (8 mg) to 4.08 (1000 mg). The test started with a filament of intermediate diameter (160 mg). If the animal showed a response to the stimulus (paw flinch or withdrawal), the next filament of a smaller diameter was applied; in the absence of response, the next filament of greater intensity was tested, and this procedure was repeated until the mechanical threshold was determined. Each filament was applied five times consecutively. The nociceptive threshold was defined as the filament of smaller diameter that induced paw withdrawal or flinches, in at least one of the five attempts [65]. The nociceptive threshold was represented as previously described [64]. 

### 4.6. Assessment of IL-6 Cytokine Levels

On the 14th day after PSNL, mice underwent the behavioral test as described above, were euthanized and the ipsilateral lumbar segment (L4-L6) of the spinal cord was collected. The samples were homogenized in 80 μL of a solution containing phosphate-buffered saline (PBS, pH 7.4) 0.4 M NaCl, Tween 100 (0.05%), 10 mM EDTA (ethylenediamine tetraacetic acid, #1610729, Bio-Rad Laboratories, São Paulo, Brazil) and protease inhibitor cocktail (1:300, Sigma-Aldrich, St. Louis, MO, USA). Next, the samples were centrifuged at 3000× *g* for 10 min at 4 °C. An aliquot of the supernatant was used to determine the protein concentration by the Bradford method [66], and samples in duplicate containing 20 µg of protein were used to determine IL-6 levels using an ELISA commercial kit according to the manufacturer’s instructions (#555240 BD OptEIA, San Diego, CA, USA). Quantification was performed using the CurveExpert 1.4 program (version 1.4.) and the detection limit was 15.6 pg/mL.

### 4.7. Cell Culture and Treatment

The BV-2 mouse microglial cells line was cultured in Dulbecco’s Modified Eagle Medium (DMEM), supplemented with 10% fetal bovine serum (FBS), streptomycin (100 µg/mL), and penicillin (100 U/mL). The BV-2 cells were seeded in 6-well plates at a density of 2 × 10^5^, at 37 °C in an incubator containing 5% CO2 and 95% air. Lipopolysaccharide (LPS) was used to induce the in vitro-activated M1 phenotype. The cells were assigned into the following groups: control (medium), LPS (100 ng/mg), LPS + 0.1 µM crotalphine, LPS + 1 µM crotalphine, and LPS + 10 µM crotalphine. For the LPS + crotalphine group, the cells were pretreated with LPS (100 ng/mg) for 24 h, the medium was removed, and then the cells were treated with different concentrations of crotalphine (0.1 µM, 1 µM, or 10 µM) for 24 h. 

### 4.8. Flow Cytometry Assay

The mean fluorescence intensity of M1 or M2 phenotype cells was determined by evaluating the level of their corresponding markers: CD86 and CD206, respectively, via flow cytometry assay. After treatment in the designed study groups, the BV-2 cells were collected and suspended in phosphate-buffered saline (PBS). Subsequently, the BV-2 cells were stained with the fluorescence-labeled CD86 (R-PE, BD Pharmingen, Franklin Lakes, NJ, USA, 09275B, 1:100) and CD206 (Alexa Fluor-647, BD Pharmingen, Franklin Lakes, NJ, USA, 565250, 1:200) antibodies for 30 min in the dark at 4 °C. FACS Canto II (Beckman Coulter, Indianapolis, IN, USA) was applied for analyzing the expression of CD86 and CD206. 

### 4.9. Statistical Analysis

The results were submitted to one-way ANOVA for ELISA and FACS analysis or two-way ANOVA for the other experiments (analysis of variance for repeated measures, followed by Tukey’s or Sidak’s post hoc tests). The GraphPad Prism software (version 6.0, La Jolla, CA, USA) was used. The significance index was given at *p* < 0.05.

## 5. Conclusions

In a conclusion, our data point out microglia as a key player in the control of chronic pain behavior induced by crotalphine, through the modulation of its state of activation, inhibiting its inflammatory M1 state and stimulating its polarization to the M2 state. In addition, our results reinforce the existence of a cross-talk between the spinal opioid and cannabinoid systems, which participate in the analgesic effect of crotalphine stimulating the release of opioid endogenous. Besides that, we highlight the role of crotalphine as an important agent for chronic pain control. 

## 6. Patents

Crotalphine is protected by the patents PI04017021, PI05023998, and BR1020160231531.

## Figures and Tables

**Figure 1 ijms-23-11571-f001:**
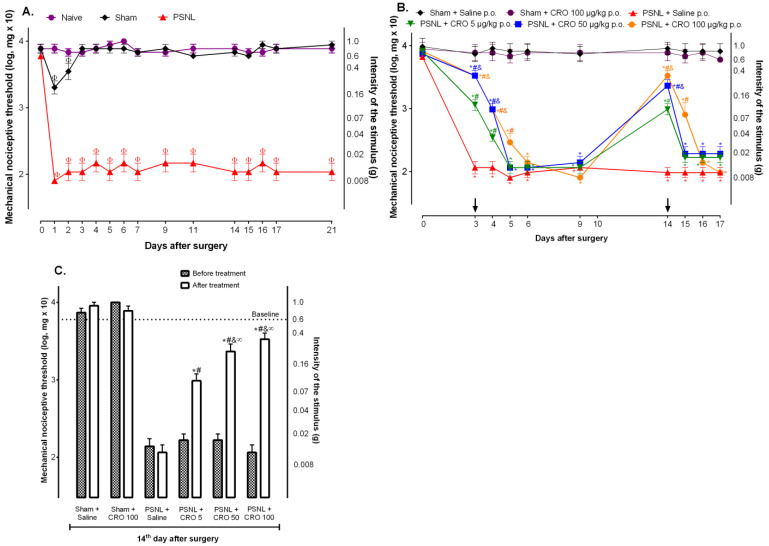
Investigation of crotalphine effect in the partial sciatic nerve injury (PSNL) model-indu ced hypernociception. Mice were submitted to surgery (sham or PSNL) and underwent behavioral testing from day 0 (baseline) to the 21st (**A**) day after surgery. Mice were treated with crotalphine (5, 50, and 100 µg/kg—CRO) (p.o.) on the 3rd and 14th (**B**,**C**) days after surgery (arrows). The nociceptive threshold was evaluated before, or 1 h after treatment (**C**), and on the following days using von Frey filaments. The results are expressed by the mean ± SEM (*n* = 5). Φ *p* < 0.05 indicates a statistical difference when compared to the Naive. * *p* < 0.05 indicates a statistical difference when compared to the Sham + Saline. # *p* < 0.05 indicates a statistical difference when compared to the PSNL + Saline. & *p* < 0.05 indicates a statistical difference when compared to the PSNL + CRO group 5 µg/kg. ∞ *p* < 0.05 indicates a statistical difference when compared before and after treatment groups. The two-way ANOVA test was used followed by Tukey’s test.

**Figure 2 ijms-23-11571-f002:**
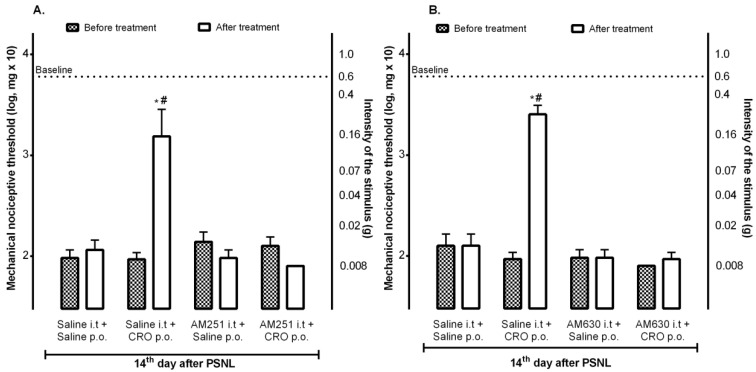
Participation of central cannabinoid receptors in the analgesic effect of crotalphine. On the 14th day after the partial sciatic nerve injury (PSNL) surgery, 10 min before crotalphine (CRO), mice were treated (i.t.) with AM251 (10 μg/10 µL) (**A**) or AM630 (2 μg/10 µL) (**B**), following the CRO (100 μg/kg, p.o.). The nociceptive threshold was evaluated using von Frey filaments one hour after CRO treatment. The results are expressed by the mean of (± EPM). *n* = 6. * *p* < 0.05 statistically significant when compared to the PSNL + Saline p.o. or PSNL + Saline i.t. + Saline p.o; # PSNL + AM251 or AM630 i.t. + CRO p.o. before and after treatment. The two-way ANOVA test was used followed by the Sidak test.

**Figure 3 ijms-23-11571-f003:**
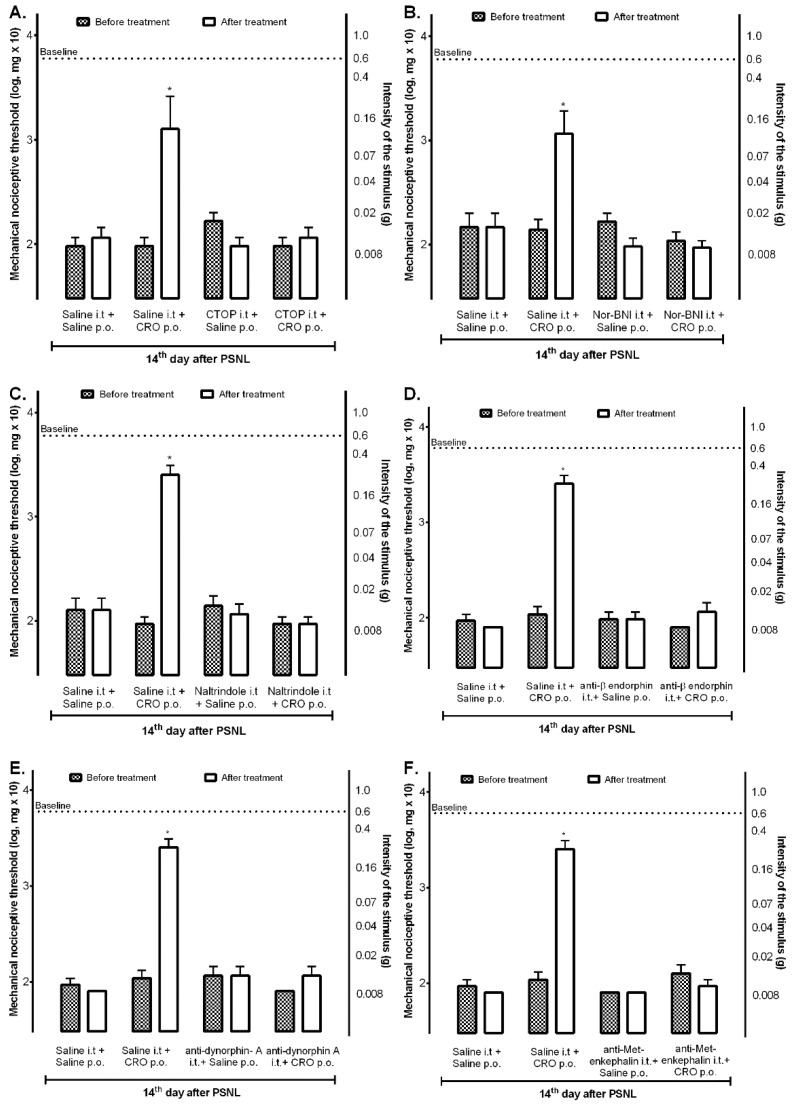
Involvement of the spinal opioid system in the crotalphine-induced analgesia. On the 14th day after the partial sciatic nerve injury (PSNL) surgery, the selective antagonists for *mu* (**A**), *kappa* (**B**), or *delta* (**C**) opioid receptors or specific antibodies for each endogenous opioid, anti-β-endorphin antibody (**D**), anti-dynorphin-A antibody (**E**) or anti-met-enkephalin antibody (**F**) were administered (i.t.). Crotalphine (CRO) (100 μg/kg, p.o.) was administered 10 min after treatments. The nociceptive threshold was evaluated using von Frey filaments 1 h after treatment with CRO. The results are expressed by the mean of ± SEM. *n* = 6. * *p* < 0.05 statistically significant when compared with the PSNL + Saline i.t. + Saline p.o. before and after treatment. The two-way ANOVA test followed by the Sidak test was used.

**Figure 4 ijms-23-11571-f004:**
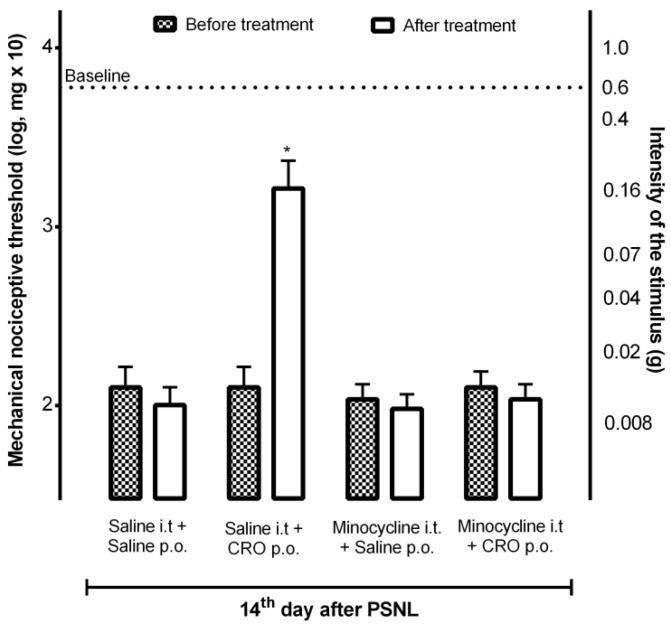
Contribution of microglia to the analgesic effect of crotalphine. Mice that underwent the partial sciatic nerve injury (PSNL) surgery, on the 14th day after surgery, were treated with minocycline i.t. and/or crotalphine (CRO) p.o. The nociceptive threshold was assessed using von Frey filaments. The results represent the group mean (±SEM). *n* = 5–6. * *p* < 0.05 indicates a statistically significant difference in relation to the PSNL + Saline i.t. + Saline p.o. The two-way ANOVA test was used followed by the Sidak test.

**Figure 5 ijms-23-11571-f005:**
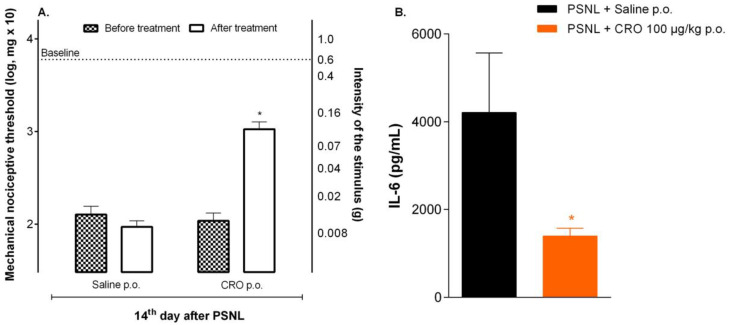
Spinal IL-6 levels after crotalphine treatment in mice underwent partial sciatic nerve injury (PSNL) surgery. On the 14th day after PSNL surgery, mice were treated with crotalphine (CRO) p.o., and after nociceptive threshold evaluation (**A**) the spinal cord was collected and processed and IL-6 levels was measured in the supernatant by enzyme immunoassay (ELISA) (**B**). Results represent the group mean (±SEM). *n* = 5–6. * *p* < 0.05 indicates a statistically significant difference in relation to the group PSNL + Salina p.o. The one-way ANOVA test was used followed by Tukey’s test.

**Figure 6 ijms-23-11571-f006:**
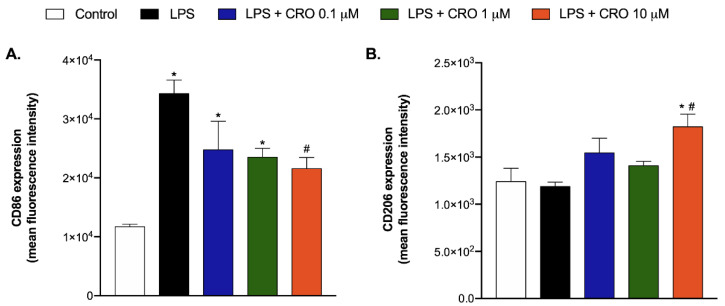
CD86 and CD206 expression in BV-2 mouse microglial cells line. BV-2 cells were treated according to the following groups: control (medium), LPS (100 ng/mg), LPS + 0.1 µM crotalphine, LPS + 1 µM crotalphine, and LPS + 10 µM crotalphine. For the LPS + crotalphine (CRO) group, the cells were pretreated with LPS for 24 h, the medium was removed, and then the cells were treated with different concentrations of crotalphine (0.1 µM, 1 µM, and 10 µM) for 24 h. The mean fluorescence intensity of CD86 (**A**) and CD206 (**B**), markers of M1 and M2 cell phenotypes, respectively, was determined via the flow cytometry assay. Results represent the group mean (±SEM). *n* = 3 assays, each one in duplicate. * *p* < 0.05 indicates a statistically significant difference in relation to the control group. # *p* < 0.05 indicates a statistically significant difference in relation to the group LPS. The one-way ANOVA test was used followed by Tukey’s test.

## Data Availability

Not applicable.

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
