# Peer review of "Crotalphine Modulates Microglia M1/M2 Phenotypes and Induces Spinal Analgesia Mediated by Opioid-Cannabinoid Systems"

_ijms, 2022, doi:10.3390/ijms231911571_

Round 1

Reviewer 1 Report

Please, find the comments in the attached file

Author Response

Reviewer 1

The manuscript by Lopes and colleagues entitled “The spinal mechanism of crotalphine analgesic effect on neuropathic pain involves microglia and opioid-cannabinoid systems activation” analyzed the effects of crotalphine on the CNS in the PSNL model of neuropathic pain. The authorsstated that crotalphine activates opioid and cannabinoid analgesic systems and modulatesthe central neuroinflammation acting on glial cells. These conclusions are based solely on the von Frey filaments test which was applied after treating the animals with crotalphine with or without specific receptor antagonists, or specific antibodies or an inhibitor of microglia activation. In my opinion, other biological experiments must be added to the manuscript in order to reach and strengthenauthors’ conclusions.

Mayor concerns

1) Does crotalphine induces toxicological effects on rats?

To the best of our knowledge, crotalphine does not induce toxicological effects. Crotalphine does not interfere with general and motor activities of rodents (doi: 10.1016/j.ejphar.2008.07.053). Even after chronic administration (administered every 3 days for 75 days), crotalphine does not induce tolerance (doi: 10.1016/j.ejphar.2008.07.053), and no changes in animal behavior and no signs of toxicological effects are visual observed. Crotalphine does not interfere with viability of DRG cells (doi: 10.3390/toxins13120912) or of human sensory-like neurons SH-SY5Y (data not published) and does not interfere with the body weight of rodents (https://doi.org/10.3390/toxins13110827). The information was included in the article.

2) Biological experiments must be performed to demonstrate the modulation of CB1 and CB2 receptors as well as mu, kappa, and delta-opioid receptors by crotalphine.

The effect of Crotalphine on CB1 and CB2 receptors as well as mu, kappa, and delta-opioid receptors was previously demonstrated (doi: 10.1111/bph.12488, doi: 10.1371/journal.pone.0090576). It was currently better explained in the text.

3) Authors must provide data that show the activation of microglia in their model. Does crotalphine induce in microglia a switch from M1 to M2 phenotypic markers?

These results were included in this new version. They were performed using microglia cell line BV2, as suggested by the other reviewer

Minor concerns

4) Avoid using direct abbreviations in the abstract

Thank you for the observation. They were corrected.

5) In the figure 1 caption, please revise the lines 110 and 111.

The Figure 1 caption was revised

6) In the figure 2 caption, please revise the lines 132 and 134.

The Figure 2 caption was revised

7) In the figure 5 B, please revise “saline PO Cro PO”

The figure was revised and corrected

8) The quality of all figures must be improved: symbols are not legible.

Sorry about it. Figures were improved

9) All abbreviations in all figure captions must be defined         

All abbreviations were defined

10) Please, revise line 327 in the material and methods section

Sorry about our mistake. The phrase was removed

Reviewer 2 Report

Review – Seongjin Lim

The spinal mechanism of crotalphine analgesic effect on neuropathic pain involves microglia and opioid-cannabinoid systems activation

They suggested that crotalphine (100 μg/kg, p.o.) induced analgesic effect was reversed by intrathecal injection of CB1 and CB2 receptor anagonists, and CTOP, nor-BNI,naltriondole and finally minocycline.

For consolidation the authors suggestion, it need to be supported Crotalphine effect on microglia cell line as BV2

In additionally, they must confirm the concentration of crotalphine in spinal cord after PO administration, since peptides do not cross BBB.

1.      Introduction commentary

-          Line 16 ‘crotalphine induced..’ -> ‘Crotalphine induced..’

-          Line 24 ‘ ..beside to activates..’ -> ‘..besides activating..’

-          Line 37 ‘..that modifies..’ -> ‘..that modify..’

-          Line 36-39 the whole sentence is too wordy and long. Punctuations are incorrect in some parts.

-          Line 39-42 ‘this condition’ refers to what? /

-          Line 41-42 ‘..which – release’ should be before ‘..including’;  otherwise, too repetitive and wordy

-          Line 44 ‘microglial cells are important players..’ -> ‘microglial cells play a significant role..’ the word important is used in the sentence before. Thus, it’s repetitive.

-          Line 46 ‘by producing and releasing’ -> ‘releasing’ too wordy & unnecessary

-          Line 48 ‘where’ -> ‘in which’

-          Line 68 ‘bone cancer pain’ -> ‘bone cancer-induced pain’

-          Line 69-70 ‘being this release dependent on’ -> ‘this release being dependent on’

-          Line 72 ‘members of the TRP cation channels superfamily’ -> ‘members of the TRP superfamily’ uncertain if ‘cation channels’ is necessary since TRP superfamily already consists of diverse groups of cation channels

-          Line 76-77  The sentence should be..‘Thus, the spinal mechanisms involved in crotalphine analgesic effects on a chronic neuropathic pain model are currently being investigated.’ Since the previous sentence is grammatically incorrect

-          Line 78-79 ‘..modulates the central neuroinflammation induced by the neuropathy, in a pathway that involves glial cells as a key player’ -> ‘modulates central neuroinflammation induced by neuropathy in a pathway that involves glial cells as key players..’

-          Line 80 ‘..of this peptide in the pain control’ -> ‘of this peptide in pain control’

-          Based on the introduction, it was difficult to indicate if the research subject is mainly about microglia or crotalphine, as there was more content about microglia than crotalphine. Information about crotalphine should be mentioned more in the introduction as it is the center of this research. Further, although it can be inferred that microglia take a huge part in the effects of crotalphine in terms of releasing analgesic substances, there must be some information about the connections between them, rather than just concisely summarizing it in the last paragraph.

-          The flow of the paragraphs doesn’t match, and they seem separated rather than being contained in the same research for their connection.

-          The sentences were generally too wordy and complicated, which may impede the readers from acquiring accurate information.

2.      Results commentary

-          It would be more accurate if the use of a specific mouse was mentioned when briefing the results instead of simply stating that ‘animals’ were used.

-          Didn’t mention the method of measuring the nociceptive threshold although written in the graph (fig. 1A)

-          Lines 92 -93 an effect cannot be performed -> ‘effect of the treatment… appeared/was expressed...’

-          Didn’t specify what the sham groups represented + merely exposing their sciatic nerve without any modifications can’t be the control group to assess the effects of crotalphin (4.2 Neuropathic pain induction – PSNL model)

-          Line 94 ‘based on the previous results 15’ – didn’t specify the unit or what the number represents

-          Lines 94-97 fig. 1B – it can’t be differentiated between the results of PSNL + CRO 100 µg/kg p.o. & PSNL + CRO 5/50 µg/kg

-          Line 99 ‘which was no longer’ -> ‘but was no longer’

-          Lines 97-100 didn’t specify which group crotalphine was administered to

-          Lines 101-102 ‘lasting in a partial way’ -> ‘lasting partially’

-          Fig. 1C contains inaccurate information – it was mentioned that sham groups weren’t conducted PSNL on, as their sciatic nerve was merely exposed and then sutured. However, the x-axis contains the sham group when it’s entitled ‘14th day after PSNL’

-          2.1: More explanation is needed in terms of how partial effects of 100 µg/kg dose were shown.

-          Lines 139-143 -> need to be more concise

-          Line 142 ‘would be also involved in’ -> ‘would also be involved in’

-          Line 144 ‘receptors antagonists’ -> ‘receptor antagonists’

-          Lines 162-166 ‘crotalphine modulates resident and inflammatory macrophages, 162 inhibiting some functions such as phagocytosis, H2O2 release, nitric oxide production, and 163 TNF-α secretion and, on the other hand, stimulating the secretion of IL-1β and modulating 164 the secretion of IL-6’ – sentence is too wordy and complicated to understand

-          Lines 169 ‘and analgesic compounds, capable to down-regulate it’ -> ‘analgesic compounds, being able to down-regulate the acticity’

-          Lines 186 -187 ‘where it expresses and release’ -> ‘ in which they express and release’

-          Lines 189 ‘ involves endogenous analgesic peptides release’ -> ‘involves the release of endogenous analgesic peptides’

-          Lines 196 – 197 (in.. observed(figure 5B) doesn’t accurately describe figure 5B

-          Taken the results altogether, it was hard to comprehend how the spinal opioid/cannabinoid receptors that mediate the analgesic effect induced by crotalphine directly relate to the microglia’s role in the expression of crotalphine’s effects. The connections between them are not explained clearly enough, so it seems as if they are two different research subjects.

-          Some of the sentences are too wordy and long, so they may cause difficulty for readers to comprehend.

3.      Discussion

-          Line 212 ‘phenomenon is a need and would help to develop more effective management strategies’ -> ‘phenomenon is needed to develop more effective pain management strategies’

-          Lines 218-222: the explanation goes from the unknown mechanism of opioid and cannabinoid system interactions in the CNS right to microglia, which lacks natural flow.

-          Lines 231 ‘crotalphine orally administered’ -> ‘crotalphine is orally administered’

-          Lines 237 ‘still longer lasting’ -> ‘still lasts longer’

-          Lines 259-261 wasn’t mentioned in the results

-          Line 265 ‘leads to the central’ -> ‘leads to central’

-          Line 266 ‘microglia have also a ‘protective state’’ -> ‘microglia are in a ‘protective state’’

Author Response

Reviewer 2

 The spinal mechanism of crotalphine analgesic effect on neuropathic pain involves microglia and opioid-cannabinoid systems activation

 They suggested that crotalphine (100 μg/kg, p.o.) induced analgesic effect was reversed by intrathecal injection of CB1 and CB2 receptor anagonists, and CTOP, nor-BNI,naltriondole and finally minocycline.

 For consolidation the authors suggestion, it need to be supported Crotalphine effect on microglia cell line as BV2

R: This assay was performed, and the results were currently included in this new version

 In additionally, they must confirm the concentration of crotalphine in spinal cord after PO administration, since peptides do not cross BBB.

R: The concentration of crotalphine in spinal cord after oral administration was not currently investigated, and it is a limitation of this study. The BBB is a selectively and permeable physical barrier, which restricts the access of a wide range of compounds into the CNS. Although it was not possible to currently determine the concentration of crotalphine at the CNS, we used analysing tools for crotalphine physiochemical properties prediction and blood-brain barrier penetrating potential, which indicates that crotalphine is not predicted to easily penetrate the blood-brain barrier. However, it is known that several disorders and pathological conditions, including neuroinflammation, disrupt the BBB integrity, increasing its permeability, allowing the transmigration of activated cells and the entrance of several compounds into the CNS. Then, it is a limitation of this study which will be future investigated. All this discussion and results of the prediction were included in this final version of the article.

  1. Introduction commentary

-          Line 16 ‘crotalphine induced..’ -> ‘Crotalphine induced..’

the text was corrected

-          Line 24 ‘ ..beside to activates..’ -> ‘..besides activating..’ ’

the text was corrected

-          Line 37 ‘..that modifies..’ -> ‘..that modify.. ’

the text was corrected

-          Line 36-39 the whole sentence is too wordy and long. Punctuations are incorrect in some parts. ’

 the text was corrected

-          Line 39-42 ‘this condition’ refers to what? /’

the text was corrected

-          Line 41-42 ‘..which – release’ should be before ‘..including’;  otherwise, too repetitive and wordy ’

the text was corrected

-          Line 44 ‘microglial cells are important players..’ -> ‘microglial cells play a significant role..’ the word important is used in the sentence before. Thus, it’s repetitive. ’

the text was corrected

-          Line 46 ‘by producing and releasing’ -> ‘releasing’ too wordy & unnecessary ’

 the text was corrected

-          Line 48 ‘where’ -> ‘in which’ ’

the text was corrected

-          Line 68 ‘bone cancer pain’ -> ‘bone cancer-induced pain’ ’

 the text was corrected

-          Line 69-70 ‘being this release dependent on’ -> ‘this release being dependent on’ ’

the text was corrected

-          Line 72 ‘members of the TRP cation channels superfamily’ -> ‘members of the TRP superfamily’ uncertain if ‘cation channels’ is necessary since TRP superfamily already consists of diverse groups of cation channels ’

the text was corrected

-          Line 76-77  The sentence should be..‘Thus, the spinal mechanisms involved in crotalphine analgesic effects on a chronic neuropathic pain model are currently being investigated.’ Since the previous sentence is grammatically incorrect ’

the text was corrected

-          Line 78-79 ‘..modulates the central neuroinflammation induced by the neuropathy, in a pathway that involves glial cells as a key player’ -> ‘modulates central neuroinflammation induced by neuropathy in a pathway that involves glial cells as key players.. ’

the text was corrected

-          Line 80 ‘..of this peptide in the pain control’ -> ‘of this peptide in pain control’ ’

the text was corrected

-          Based on the introduction, it was difficult to indicate if the research subject is mainly about microglia or crotalphine, as there was more content about microglia than crotalphine. Information about crotalphine should be mentioned more in the introduction as it is the center of this research. Further, although it can be inferred that microglia take a huge part in the effects of crotalphine in terms of releasing analgesic substances, there must be some information about the connections between them, rather than just concisely summarizing it in the last paragraph.

The text was changes according the suggestions

-          The flow of the paragraphs doesn’t match, and they seem separated rather than being contained in the same research for their connection.

Paragraphs were connected

-          The sentences were generally too wordy and complicated, which may impede the readers from acquiring accurate information.

  Sorry about that. The text was changed according the suggestions

  1. Results commentary

-          It would be more accurate if the use of a specific mouse was mentioned when briefing the results instead of simply stating that ‘animals’ were used.

Thank you for the suggestion. The words were replaced.

-          Didn’t mention the method of measuring the nociceptive threshold although written in the graph (fig. 1A).

The method used was included in the results, in addition to the description in the methods section.

-          Lines 92 -93 an effect cannot be performed -> ‘effect of the treatment… appeared/was expressed...’

You are completely right. The term was corrected

-          Didn’t specify what the sham groups represented + merely exposing their sciatic nerve without any modifications can’t be the control group to assess the effects of crotalphin (4.2 Neuropathic pain induction – PSNL model)

Both sham and naïve rats were used as controls. This information is cited in the item 4.2 and the groups were shown in the figure 1

-          Line 94 ‘based on the previous results 15’ – didn’t specify the unit or what the number represents

Sorry. The number represents a bibliographic citation. It was corrected.

-          Lines 94-97 fig. 1B – it can’t be differentiated between the results of PSNL + CRO 100 µg/kg p.o. & PSNL + CRO 5/50 µg/kg.

The description in the text was rewritten. At the graphic, it is represented at the day 5 (48 h after the administration which occurred at the day 3), where partial analgesia remains only at the dose of 100 ug/kg.

-          Line 99 ‘which was no longer’ -> ‘but was no longer’

the text was corrected

-          Lines 97-100 didn’t specify which group crotalphine was administered to

the text was corrected

-          Lines 101-102 ‘lasting in a partial way’ -> ‘lasting partially’

the text was corrected

-          Fig. 1C contains inaccurate information – it was mentioned that sham groups weren’t conducted PSNL on, as their sciatic nerve was merely exposed and then sutured. However, the x-axis contains the sham group when it’s entitled ‘14th day after PSNL’

The x-axis was corrected

-          2.1: More explanation is needed in terms of how partial effects of 100 µg/kg dose were shown.

The partial reversion is defined once the value obtained in the group is statistically different from the group PSNL and also different from the group sham. The symbols in the graphics were improved in order to better visualization.

-          Lines 139-143 -> need to be more concise the text was corrected

-          Line 142 ‘would be also involved in’ -> ‘would also be involved in’ the text was corrected

-          Line 144 ‘receptors antagonists’ -> ‘receptor antagonists’

the text was corrected

-          Lines 162-166 ‘crotalphine modulates resident and inflammatory macrophages, 162 inhibiting some functions such as phagocytosis, H2O2 release, nitric oxide production, and 163 TNF-α secretion and, on the other hand, stimulating the secretion of IL-1β and modulating 164 the secretion of IL-6’ – sentence is too wordy and complicated to understand –

the text was changed

-          Lines 169 ‘and analgesic compounds, capable to down-regulate it’ -> ‘analgesic compounds, being able to down-regulate the acticity’

the text was corrected

-          Lines 186 -187 ‘where it expresses and release’ -> ‘ in which they express and release’

 the text was corrected

-          Lines 189 ‘ involves endogenous analgesic peptides release’ -> ‘involves the release of endogenous analgesic peptides’

the text was corrected

-          Lines 196 – 197 (in.. observed(figure 5B) doesn’t accurately describe figure 5B

The text was rewritten

-          Taken the results altogether, it was hard to comprehend how the spinal opioid/cannabinoid receptors that mediate the analgesic effect induced by crotalphine directly relate to the microglia’s role in the expression of crotalphine’s effects. The connections between them are not explained clearly enough, so it seems as if they are two different research subjects

Activated microglia is responsible for the production and release of analgesic compounds, such as endorphins and endocannabinoids, both in vivo and in vitro, thus regulating pain transmission. Both opioid and cannabinoid compounds are known to be effective analgesics by reducing inflammation and excitability of neuronal cells leading to decreased transmission of nerve impulses alongside inhibition of neurotransmitter release. This information is stated at introduction. We hope that this information, together with all the other changed made in the text would clarify this point.

-          Some of the sentences are too wordy and long, so they may cause difficulty for readers to comprehend.

   Sorry about that. The text was changed according the suggestions

  1. Discussion

 -          Line 212 ‘phenomenon is a need and would help to develop more effective management strategies’ -> ‘phenomenon is needed to develop more effective pain management strategies’

the text was corrected

-          Lines 218-222: the explanation goes from the unknown mechanism of opioid and cannabinoid system interactions in the CNS right to microglia, which lacks natural flow.

Thank you for the advertising. The text was changed

-          Lines 231 ‘crotalphine orally administered’ -> ‘crotalphine is orally administered’

the text was corrected

-          Lines 237 ‘still longer lasting’ -> ‘still lasts longer

the text was corrected

-          Lines 259-261 wasn’t mentioned in the results

the text was corrected

-          Line 265 ‘leads to the central’ -> ‘leads to central

the text was corrected

-          Line 266 ‘microglia have also a ‘protective state’’ -> ‘microglia are in a ‘protective state’

the text was corrected

Round 2

Reviewer 1 Report

.

Author Response

After revision, the quality and the clarity of the manuscript have improved.

R: Thank you very much

However, some points remain to be addressed.

  • In the paper https://doi.org/10.1111/bph.12488 (figure3) it was demonstrated that Crotalphine induces peripheral activation of opioid and cannabinoid receptors using conformation statesensitive antibodies in the presence or not of specific receptor antagonists in the skin tissue. In my opinion, data that show that crotalphine activates the opioid and cannabinoid receptors in the spinal cord tissue of PNSL mice are necessary to support the von Frey filaments test results.

R: We agree that evaluating the activation of opioid receptors at the SNC would improve this manuscript. Unfortunately, we do not have samples (intact spinal cord tissue from these animals) to perform this test, since the spinal cords were processed for ELISA assay. In fact, crotalphine induces opioid and cannabinoid receptors activation, as previously demonstrated in the article cited above. The previous data, together with the current results showing that opioid and cannabinoid antagonists, as well as antibodies against endogenous opioids, block crotalphine-induced analgesia, confirm the involvement of these systems in the peptide effects. Importantly, considering that there is no structural difference between peripheral and central opioid receptors, we expect that crotalphine activates these receptors, directly or indirectly. We added a sentence in the Discussion stating that this evaluation is necessary and will be further investigated.

  • Based on the results shown, I suggest to soft the title of the manuscript since the modulation of microglia M1/M2 states was only demonstrated in BV-2 mouse microglial cell line and not in the spinal cord PNSL model. Moreover, not only microglia but also neurons are sources of IL-6 in the spinal cord. Therefore, the modulation of microglia M1/M2 states is in spinal cord of PNSL mice must the added to support the current title.

R: The title was changed

  • Finally, I suggest adding in the figure 6 a representative plot of flow cytometer data.

R: The histograms from flow cytometry were included as supplementary material

R: Thank you very much for your comments and suggestion, that really improved our article. We hope it is suitable now for publication. 

Reviewer 2 Report

I have no furtther comments

Author Response

Thank you very much for your comments and dedication in the review of this article. We are sure that your suggestions really improved our work. 

Round 3

Reviewer 1 Report

.